# Relationship Between Estimated Drug Distribution of Antiretroviral Therapy and Immune Proteins in Cerebrospinal Fluid During Chronic HIV Suppression

**DOI:** 10.3390/v17060749

**Published:** 2025-05-23

**Authors:** Mattia Trunfio, Jennifer E. Iudicello, Patricia K. Riggs, Asha R. Kallianpur, Todd Hulgan, Ronald J. Ellis, Scott L. Letendre

**Affiliations:** 1HIV Neurobehavioral Research Program, Departments of Neurosciences and Psychiatry, University of California, 220 Dickinson St., San Diego, CA 92103, USA; mtrunfio@health.ucsd.edu (M.T.); jiudicello@health.ucsd.edu (J.E.I.); pariggs@health.ucsd.edu (P.K.R.); roellis@health.ucsd.edu (R.J.E.); 2Division of Infectious Diseases and Global Health, University of California, 9500 Gilman Dr., La Jolla, CA 92093, USA; 3Department of Genomic Medicine, Cleveland Clinic Lerner Research Institute, Cleveland, OH 44106, USA; kalliaa@ccf.org; 4Department of Medicine, Vanderbilt University Medical Center, Nashville, TN 37232, USA; todd.hulgan@vumc.org

**Keywords:** chronic neuroinflammation, neuro HIV, central nervous system penetration efficacy, CPE, IL-6, CXCL10, TNF-α, antiretrovirals, neurotargeted antiretroviral therapy

## Abstract

Antiretroviral therapy (ART) drugs vary in their distribution into cerebrospinal fluid (CSF), which can be estimated using the central nervous system (CNS) penetration effectiveness (CPE) score. Although higher CPE has been associated with lower CSF HIV RNA levels, its relationship to CSF inflammation is less clear. We investigated associations between CPE and three CSF immune biomarkers (CXCL10, TNF-α, and IL-6) in 275 virally suppressed people with HIV (PWH) on three-drug ART regimens using a training–validation design. Participants were randomized into training (TG, n = 144) and validation (VG, n = 131) groups with similar demographics, ART characteristics, and CPE scores. The CSF levels of the biomarkers were quantified by bead suspension array-based immunoassays. In both groups, higher CPE correlated with lower levels of CXCL10 (TG: r = −0.31, *p* < 0.001; VG: r = −0.30, *p* < 0.001) and TNF-α (TG: r = −0.19, *p* = 0.04; VG: r = −0.18, *p* = 0.03), with remarkably similar effect size. CPE did not correlate with IL-6 in either group. Multivariable models confirmed the associations between higher CPE and both lower CXCL10 (R^2^ = 0.16, *p* < 0.001) and TNF-α (R^2^ = 0.07, *p* = 0.02) in CSF, and supported the relative resistance of IL-6 to ART effects. During suppressive ART, regimens that achieve higher concentrations in the CNS may better reduce some indicators of CSF inflammation (CXCL10 and TNF-α, closely related to the interferon pathway), but they may not fully normalize the neuroimmune environment (IL-6). Distinct ART regimens may produce different neuroimmune signatures, potentially contributing to heterogeneous patterns of brain injury.

## 1. Introduction

The central nervous system (CNS) penetration effectiveness (CPE) score was developed to estimate how well antiretroviral therapy (ART) drugs distribute into the CNS using pharmacokinetic and pharmacodynamic data [1], and thus how well they control HIV replication in this compartment. For instance, higher CPE scores have been associated with lower HIV RNA levels in the cerebrospinal fluid (CSF), suggesting more effective viral suppression in the CNS [2,3], and adjusting CPE to account for an individual’s cumulative drug resistance profile improves its predictive value for CSF viral escape [4]. Yet, it remains unclear whether higher estimated ART drug distribution into the CNS (eADDC) mitigates neuroinflammation. Persistent neuroinflammation during long-term viral suppression is increasingly recognized as a driver of neurocognitive and psychiatric complications in PWH [5,6]. While CPE is a practical, scalable proxy measure of eADDC and viral control in the CNS, neuroinflammation may persist due to HIV transcription, protein production, or residual replication in brain tissues [7], even when CSF HIV RNA is undetectable by standard assays. As the relationship between CPE and neuroinflammatory markers has not been systematically examined in virally suppressed individuals, a critical gap remains regarding whether metrics of eADDC can also be informative tools to estimate CNS inflammation and immune activation.

In recent years, analyses of eADDC and CPE have slowed, in part because studies comparing estimates of drug distribution into the CNS to clinical outcomes had conflicting results. Some studies found higher CPE to be associated with better cognition, others with worse [8,9,10], and clinical trials that modified ART regimens based on eADDC found no cognitive benefit [11,12,13]. While some of the unmeasured confounders behind these discrepancies have been mitigated by the introduction of newer drugs and the earlier initiation of ART (e.g., reduced antiretroviral -ARV -neurotoxicity, legacy effects [14,15]), the knowledge gap regarding the relationship between eADDC and neuroinflammation remains a key limitation in interpreting these conflicting findings.

Independently of their antiviral activity and the consequences of viral suppression, ARVs can exert class- and molecule-specific direct effects on immune cells, for example, tenofovir/emtricitabine stimulation of the interferon (IFN) I/III pathway [16], abacavir stimulation of the inflammasome [17], NRTI activation of the Wnt5a signaling pathway in the CNS [18], INSTIs’ inability to reverse respiratory and mitochondrial dysfunction in CD4+ T cells [19], and PIs’ immunomodulatory effects on lymphocytes [20]. Similarly, ARVs also have direct effects on CNS cells, such as PIs altering amyloid precursor protein processing [21], the involvement of darunavir and ritonavir in tau and microglia pathology [22], and NRTIs and INSTIs causing mitochondrial dysfunction and neuronal toxicity [23,24]. Emerging evidence further suggests differences between PIs and NNRTIs in their efficacy in suppressing residual replication and viral transcription in the blood [25]. Such differences in “residual viral control” may vary even more in the CNS, as intracellular penetration and inhibitory concentrations of ARVs vary across CNS cell types to a greater extent than in PBMCs [26,27,28]. Thus, greater eADDC (higher CPE) is associated with more effective viral suppression, but whether greater eADDC is also associated with lower neuroinflammation in the setting of chronic viral suppression remains controversial.

We addressed this gap using a cross-sectional training–validation design to test whether higher CPE scores are associated with lower levels of three key CSF biomarkers of inflammation—CXCL10, TNF-α, and IL-6—in a large, well-characterized cohort of PWH with long-term suppressed plasma and CSF HIV RNA.

## 2. Materials and Methods

The CNS HIV Antiretroviral Therapy Effects Research (CHARTER) study is an ongoing cohort study in which PWH are followed, with regular evaluations that include lumbar puncture for research purposes. Participants were selected for this analysis if they were treated with a three-drug ART regimen for at least 6 months, had HIV RNA level ≤50 copies/mL in the CSF and plasma, and had Tumor Necrosis Factor-alpha (TNF-α), Interleukin 6 (IL-6), and C-X-C Motif Chemokine Ligand 10 (CXCL10) levels measured in the CSF.

Participants with untreated systemic or CNS infections, or other potentially confounding neurological conditions (e.g., stroke, head trauma, CNS cancers, CNS autoimmune diseases, epilepsy, and psychiatric disorders other than anxiety and depression, such as schizophrenia) were excluded to reduce the impact of unrelated neurological comorbidities. These exclusions limit generalizability but strengthen the internal validity of our findings by allowing a more specific focus on neuroinflammation in the context of HIV and ART. The resulting 275 participants were randomly divided into two groups: the training/discovery group (TG) and the validation group (VG).

The protocol was approved by the Institutional Review Board at the University of California in San Diego, CA, USA, and written consent was obtained from all participants.

The three biomarkers are produced by activated immune cells, including B and T lymphocytes, and have pleiotropic effects, such as the stimulation and migration of activated monocytes and lymphocytes, and interferon-induced antiviral response [29,30,31]. All have been found to be persistently elevated in the blood and CSF despite viral suppression [32], and have been variably associated with brain health in PWH (cognitive impairment [33] and depression [5,34]).

Biomarkers were measured in pg/mL in CSF stored at −80 °C via suspension bead arrays (Luminex FlexMap 3D platform) with commercially available kits (EMD Millipore, Billerica, MA, USA). CSF samples were collected between 2003 and 2008 via morning spinal taps to minimize diurnal variability in CSF concentrations. All the assays were performed at the same laboratory with the same instrument. CPE score was calculated as previously described [1]; briefly, each ART drug is assigned a score based on its pharmacokinetics (e.g., CSF concentration), physicochemical properties (e.g., lipophilicity), and known CNS efficacy from clinical studies. The total CPE score for an individual is the sum of each drug score within the regimen, with higher scores indicating better CNS penetration and potential efficacy.

Non-normally distributed variables (e.g., biomarkers) were log10-transformed to reduce skewness. Data were summarized with mean (standard deviation) or median (interquartile range) for continuous variables and number (%) for categorical variables. To assess whether continuous covariates were appropriately modeled as linear predictors, we applied multivariable fractional polynomial regression and assessed LOESS curves. The TG and VG were compared using two-sided independent *t*-tests, Fisher’s exact tests, or Mann–Whitney U tests, as appropriate. Pearson’s correlations were performed in both the groups to evaluate the relationships between CPE scores and neuroinflammatory biomarkers. Multivariable linear regression models were used to assess the independent associations between CPE and CSF biomarker levels. Two model-building strategies were employed, and were run on the whole study population to increase the power and allow multiple adjustments. Model 1 included all covariates with significant univariable associations (*p* < 0.05) and retained them without further selection; to ensure robustness and maintain the internal validation approach, these models were also bootstrapped using 1000 random samples. Model 2 included the same univariable-significant covariates, along with the predefined confounders (age, sex, race, and ART duration), and then applied backward selection based on the Akaike information criterion (AIC). Due to the nature of backward selection, bootstrapping was not applied to Model 2. Biomarker values were log_10_-transformed prior to modeling, and regression coefficients are reported on the log_10_-transformed scale (i.e., log_10_ pg/mL). Exploratory analyses on the associations between single ARVs and CSF biomarkers levels were also performed with the same modeling approach. JMP v.17 (SAS Institute, Cary, NC, USA) was used for the analyses.

## 3. Results

A total of 275 participants were assessed. The randomization procedure assigned 144 participants to the TG and 131 to the VG.

As shown in Table 1, baseline demographic and disease characteristics were similar between the two groups, including CSF levels of the three biomarkers and the CPE score (7.0 [range 4–10] and 7.1 [range 3–10] in TG and VG, *p* = 0.96).

The mean age was 43.8 years in both groups (*p* = 0.94); most participants were male, and approximately half were white. The groups had a similar estimated duration of HIV infection, and although about two-thirds of participants carried an AIDS diagnosis, the median CD4+ T-cell count was >500 cells/µL at the time of inclusion (Table 1).

The most common ART backbones were emtricitabine/tenofovir (39.6%), lamivudine/zidovudine (17.4%), and lamivudine/tenofovir (10.2%), while the most common third drugs were efavirenz (34.5%), atazanavir/ritonavir (26.9%), nevirapine (14.9%), and lopinavir/ritonavir (12.7%; Table 1); no differences in the composition of ART regimens were observed between the TG and VG groups (comparisons for the prevalent backbones and third-drug data are shown in Table 1). Seventy-eight participants (28.4%) were on their first ART regimen, and the duration of both current and lifetime ART regimens was similar (Table 1).

Higher CPE values correlated with lower CSF CXCL10 and CSF TNF-α in both the TG and VG, with almost identical correlation coefficients (Figure 1). In contrast, CPE did not correlate with CSF IL-6 in either the TG or the VG (Figure 1). CSF CXCL10 and CSF TNF-α levels positively correlated with each other in both the TG (r = 0.704, *p* < 0.001) and VG (r = 0.626, *p* < 0.001), while neither was correlated with CSF IL-6 in either group.

Since the groups did not differ, they were then combined to increase power in the multivariable analyses. Bootstrapping (1000 samples) was applied to all models to maintain the internal validation approach. In this larger sample, the significant relationship between higher CPE and lower CSF TNF-α was confirmed via univariable analysis (Appendix A) and in both multivariable models (Table 2). Higher CPE also remained associated with lower CSF CXCL10 in the univariable analysis (Appendix A) and in both multivariable models (Table 2). No demographic, clinical, or HIV-related characteristics were associated with CSF IL-6 levels in the univariate analysis (Appendix A).

To further investigate the relationship between individual ARVs and CSF inflammation, we conducted exploratory analyses based on the most commonly used ARVs, as reported in Table 1. Univariable associations between single ARVs and CSF biomarkers (Appendix A) revealed that drugs with a lower CPE ranking (tenofovir disoproxil and atazanavir, both CPE ≤ 2) were associated with higher levels of CSF CXCL10 and TNF-α. Conversely, ARVs with the highest CPE (zidovudine and nevirapine, both CPE = 4) were associated with lower levels of these biomarkers. Among drugs with intermediate CPE (CPE = 3), only lopinavir showed a significant association with lower TNF-α levels (Appendix A).

In multivariable models for CSF CXCL10 (Appendix A), CPE remained independently associated with lower biomarker levels, even after the inclusion of individual ARVs. Notably, atazanavir and tenofovir disoproxil were associated with higher CXCL10 only in models excluding CPE. For CSF TNF-α (Appendix A), CPE and all five univariable-selected ARVs were not significantly associated in the full models. However, when CPE was excluded, atazanavir showed significant associations with higher TNF-α levels and lopinavir with lower biomarker levels, while backward selection retained only atazanavir as associated with higher TNF-α levels (Appendix A). Across all models, collinearity was detected between CPE and ARVs (VIFs ranging from 1.2 to 4.5), and models excluding CPE explained mildly less variance (lower R^2^).

## 4. Discussion

The persistence of systemic and CNS inflammation during HIV infection despite ART has been associated with mortality and the burden of comorbidities [35]. While a cure for HIV remains the ultimate goal, the continuing elusiveness of this goal means that a better understanding of inflammation in PWH is needed.

In the current study, we examined three inflammation biomarkers in CSF that have been associated with brain health disorders in PWH, including neurocognitive impairment and mood disorders [5,36,37,38]. Two (CXCL10 and TNF-α) correlated inversely with eADDC. While the effect size of such associations was small, the values were strikingly similar in the TG and VG, and confirmed in adjusted models computed using 1000 random sub-samples; moreover, while covariates other than CPE were independently associated with CSF CXCL10 and TNF-α, including sex and AIDS diagnosis, their effect size was similar to that of CPE. These findings suggest that inflammation in CSF in PWH is at least partially dependent on CSF ART drug concentrations. This could occur via the suppression of HIV replication, though we acknowledge that this study does not provide definitive evidence of this since all participants had plasma and CSF HIV RNA levels < 50 copies/mL. In line with past studies, where plasma CXCL10 and other inflammatory biomarkers did correlate with HIV RNA [39,40], it is also possible that a more sensitive single-copy HIV quantification assay could have shown that the relationship between CSF CXCL10, TNF-α, and CPE is dependent on a relationship between CPE and CSF HIV RNA in the narrow range of 1–49 copies/mL.

CSF levels of CXCL10 and TNF-α were positively correlated, whereas IL-6 did not correlate with either. This finding aligns with the known interplay between CXCL10 and TNF-α within the IFN signaling pathway, as described in other disease models involving chronic inflammation [41,42,43,44]. TNF-α can induce the production of IFN-γ, which subsequently stimulates CXCL10 expression in various CNS cells, including astrocytes and microglia [44,45,46]. IFN-γ is highly sensitive to viral RNA, as it is induced downstream of innate immune activation by pattern recognition receptors such as Toll-like receptor 7 and 8, as well as cytosolic RNA sensors, which detect HIV RNA and trigger cytokine signaling, including TNF-α and CXCL10 production [47]. Compared to CXCL10 and TNF-α, IL-6 is less tightly associated with IFN pathways: the production of IL-6 in the CNS is multifactorial, involving various cell types such as microglia, astrocytes, and neurons, and can be triggered by multiple stimuli, including non-viral factors and other pro-inflammatory cytokines [48,49]. Furthermore, IL-6 can activate the JAK/STAT3 signaling pathway, leading to its sustained production [50]. This persistent expression may be influenced by factors beyond viral load, such as neuronal injury or glial cell activation [48], which are not fully addressed by ART. Thus, the absence of an association between CPE and CSF IL-6 could be due to the relatively stable expression of IL-6 due to the presence of alternative inflammatory signals, even in the absence of viral replication. In this regard, no demographic or clinical factors analyzed in the study were associated with CSF IL-6 levels. While multiple studies have shown that IL-6 levels decrease during ART, evidence also supports that its decay in PWH is slower than that of other inflammation biomarkers, and on average will never reach concentrations observed in people without HIV [51,52,53]. Overall, IL-6 may be a better marker of legacy effects (e.g., residual immunological alterations and CNS injury established before ART initiation and that persist despite long-term viral suppression [54]). As such, IL-6 may reflect cumulative inflammatory burden rather than being a sensitive indicator of current or residual viral activity, explaining our negative findings, as well as previous observations [51,52,53]. Lastly, while several studies have reported association between higher IL-6 in the blood and poorer cognitive performance in PWH [32,55,56], in the CSF, lower levels of IL-6 have been consistently associated with cognitive impairment [33]. Thus, the absence of a negative correlation between CPE and CSF IL-6, as found for CXCL10 and TNF-α, is arguably less concerning than if higher CPE had been associated with lower IL-6 levels.

Modification of the pathways of all three biomarkers investigated could be beneficial, and is already under investigation in populations with and without HIV infection [29,30,31]. For example, clinical trials on ruxolitinib, a JAK 1/2 inhibitor with IL-6-modulating properties, have been initiated through the AIDS Clinical Trials Group [57]. Furthermore, several specific co-medications (e.g., antidepressants, statins) have also shown modulating effects on specific cytokines and chemokines in the blood and CSF of PWH [58,59,60].

The cross-sectional design did not allow us to evaluate the relationships over time, but another explanation of our findings could be that higher CPE could also measure the stability or continuity in effective viral suppression over time: lower CPE scores may allow viral blips in the CNS, which in turn could stimulate transient immune responses, inflammation, and tissue injury [61,62]. Both explanations are possible and supported by data on ART adherence, plasma blips, low-level viremia, and neurocognitive outcomes [39,63,64,65], and by the higher risk of both blips and viral escape in the CSF of PWH on ART with CPE scores that decrease after adjustment for historical drug resistance mutation profiles [4,66].

Even though we adjusted for the duration of ART, sex, race, and age, we likely did not fully account for all confounding factors that might influence neuroinflammation, such as co-infections other than HCV [67,68], genetic predisposition [69,70], comorbidities and co-medications (e.g., antidepressants, statins) [58,71,72], and interindividual differences in ART metabolism and HIV reservoirs [73,74]. For example, while more transcriptionally active reservoirs are associated with higher inflammation [75], no data are available to definitively answer whether differences in eADDC over time affect the decay of the HIV reservoir in the brain. The CPE score in our analysis referred to the ART regimen at the time of CSF collection, and participants had been on that regimen for an average duration of one year. The reservoir decay that follows ART initiation is more pronounced in the first year, and thereafter the decay kinetics weaken [76]. About one-fourth of the participants were on their first ART regimen, and the hypothesis of an association between CPE and inflammation mediated by reservoir decay may fit this subgroup. We do not know whether the other participants started the study regimen after viral failure or for other reasons and under continuous viral suppression yet the association between CPE and biomarkers was independent from the duration of current and lifetime ART. Longitudinal studies are required to evaluate whether CPE may be informative in predicting the dynamics of both inflammation and HIV reservoirs in the CNS. Assessing cumulative eADDC/CPE across an individual’s lifetime ART regimens may be more informative for long-term CNS outcomes, such as neurocognitive decline or biomarkers that reflect more chronic or multifactorial processes (e.g., IL-6), rather than those that change rapidly and significantly in response to short-term triggers. Up-to-date data in this regard remain limited and conflicting: a study by the Neurocognitive Assessment in the Metabolic and Aging Cohort found no significant association between cumulative CPE scores and neurocognitive impairment in well-treated PWH [8]. Conversely, a case–control study in Taiwan reported that high cumulative CPE scores were associated with an increased risk of neurological diseases in PWH with low adherence, while low cumulative CPE scores were protective against cognitive impairment in adherent PWH [41]. Contrary to our findings, these suggest that cumulative CPE may be an indicator more sensitive to ART neurotoxic effects rather than the degree of viral suppression in the CNS.

Due to the large variety of factors that are influenced by ARVs and can, in turn, influence ARV effects (e.g., blood–brain barrier impairment, presence and relative abundance of cell types, co-medications and drug–drug interactions, aging, metabolic and physiologic traits) [26,70,77,78,79,80], different ART regimens may modify the neuroimmune environment to varying degrees and through different effects. In line with this, we conducted exploratory analyses of the single ARVs most commonly used in our cohort to assess whether specific agents might be independently associated with CSF inflammation. In univariable models, drugs with lower CPE were associated with higher levels of CXCL10 and TNF-α, while those with the highest CPE score were associated with lower biomarker levels. This pattern suggests a potential gradient of effect, indicating that CNS inflammation may be more closely tied to drug penetration than to molecule-specific effects. In the multivariable models, CPE remained significantly associated with CXCL10, even after adjusting for individual ARVs. The inclusion or exclusion of CPE influenced the associations of atazanavir and tenofovir with CXCL10, with both ARVs associated with higher CSF levels when CPE was excluded. For TNF-α, the significance of CPE, atazanavir, and lopinavir varied depending on whether they were included together in the models. These findings are exploratory and should be interpreted cautiously as limited by sample size, model complexity, collinearity issues, heterogeneous comparators, and shared explanatory variance.

These complexities also underscore the broader challenge that, even beyond individual ARVs, ART regimens as a whole may not fully normalize the neuroimmune environment. As for peripheral inflammation, ART regimens may not fully normalize the neuroimmune environment, which could lead to different patterns of brain injury being associated with different ART regimens. For example, compared to the older regimens included in this study, modern ART is often based on INSTIs. While INSTIs are associated with potent viral suppression, improved tolerability, and reduced systemic inflammation, they can also cause neuropsychiatric side effects [81] and exert neurotoxic effects (e.g., dysregulation of iron transport and mitochondrial dysfunction in HIV-infected microglia and neural-lineage cells [24]). Although INSTIs generally have moderate-to-high eADDC, their intracellular penetration across different CNS cell types varies as much as for other ARV classes [27,28]. Ultimately, the net impact of the sum of INSTIs’ effects contributing to and mitigating neuroinflammation remains unclear, as is also the case for second-generation NNRTIs, long-acting injectable regimens, and newly licensed ARV classes (e.g., capsid and attachment inhibitors). Future studies should assess whether these newer regimens provide greater neuroprotection or, conversely, introduce novel CNS-related adverse effects that could influence brain health in people with HIV.

With recent changes in prescribing practices, new data on ART drug concentrations in CSF and brain tissue [78,82,83], and the rollout of new ARV classes, it is crucial to reassess how eADDC relates to CNS outcomes in this evolving landscape. Further research is needed to better define the relationship between persistent inflammation and ART during HIV infection, and to transition from CNS-tailored ART regimens aimed solely at viral suppression [84] to regimens designed to achieve CNS viral control while not fueling neuroinflammation, potentially promoting beneficial neuroimmune modulation. Accordingly, a deeper understanding of which biomarkers and pathways are more sensitive or refractory to the effects of specific ARV classes and molecules may lead to tailoring ART regimens to address chronic inflammation once sustained viral suppression has been achieved. While our findings suggest that CPE may be a valuable tool for investigating these questions in a hard-to-reach and delicate sanctuary such as the CNS, none of the newest ARVs were evaluated or included in this score, limiting the ability to reproduce our findings in more modern cohorts of PWH.

## Figures and Tables

**Figure 1 viruses-17-00749-f001:**
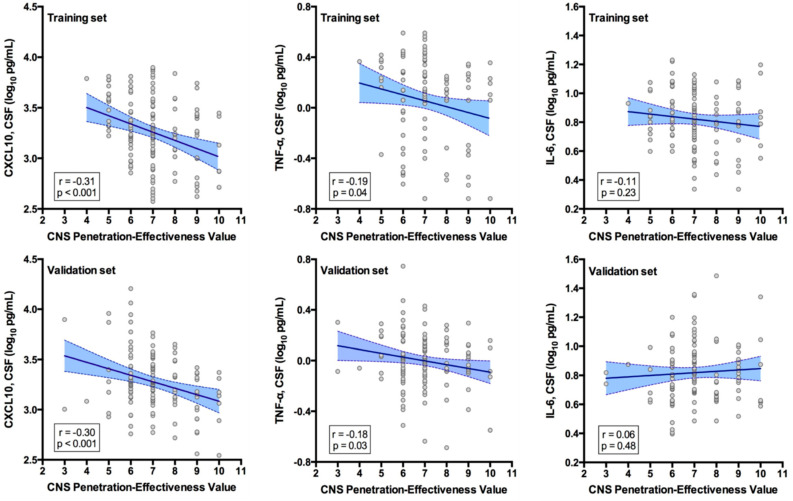
Pearson’s correlations between CPE and CSF biomarkers in the training and validation groups. The **upper rows** of the graphs show the correlations between CPE and CXCL10 (**left panel**), TNF-α (**middle panel**), and IL-6 (**right panel**) within the training/discovery group (n = 144). The **lower rows** of the graph show the correlations between CPE and CXCL10 (**left panel**), TNF-α (**middle panel**), and IL-6 (**right panel**) within the validation group (n = 131). Higher CPE values significantly correlated with lower levels of CXCL10 and TNF-α in the CSF of both groups, with remarkably almost identical correlation coefficients. In contrast, CPE did not correlate with CSF IL-6 in either the training or the validation group. Legend: CXCL10 = C-X-C Motif Chemokine Ligand 10; CSF = cerebrospinal fluid; TNF-α = Tumor Necrosis Factor alpha; IL-6 = Interleukin 6.

**Table 1 viruses-17-00749-t001:** Demographic and clinical characteristics of training group and validation group.

Characteristics	Training Group (n = 144)	Validation Group (n = 131)	*p* Value
Age, years	43.8 (±8.3)	43.8 (±8.0)	0.94
Male sex, n	122 (84.7%)	101 (77.1%)	0.11
White ethnicity, n	72 (50.0%)	57 (43.5%)	0.28
Body mass index	26.1 (±5.3)	26.4 (±4.5)	0.56
HCV seropositive, n	23 (16.0%)	49 (37.4%)	<0.001
Duration of HIV, years	10.4 (3.6–14.8)	9.5 (4.9–15.0)	0.20
AIDS diagnosis, n	94 (65.3%)	83 (63.3%)	0.74
Current CD4+ count, cells/µL	531 (310–692)	532 (357–691)	0.97
CD4/CD8 ratio	0.58 (0.38–0.91)	0.61 (0.37–0.90)	0.82
Nadir CD4+ count, cells/µL	167 (32–266)	180 (49–265)	0.46
Current ART regimen duration, months	9.2 (5.0–21.3)	13.0 (4.6–29.8)	0.36
Lifetime ART regimen duration, months	47.7 (16.4–76.9)	54.2 (20.7–90.0)	0.86
ART backbone, n			
FTC/TDF	60 (41.7%)	49 (37.4%)	0.47
3TC/ZDV	21 (14.6%)	27 (20.6%)	0.19
3TC/TDF	15 (10.4%)	13 (9.9%)	0.89
3TC/ABV	12 (8.3%)	8 (6.1%)	0.48
DDI/TDF	6 (4.2%)	9 (6.9%)	0.32
ABV/TDF	5 (3.5%)	4 (3.1%)	0.84
3TC/D4T	3 (2.1%)	2 (1.5%)	0.73
Other NRTI combinations	15 (10.4%)	12 (9.2%)	0.73
3 ARV classes	7 (4.9%)	7 (5.3%)	0.86
Prevalent third antiretrovirals, n			
EFV	47 (32.6%)	48 (36.6%)	0.49
ATV/r	44 (30.6%)	30 (22.9%)	0.15
NVP	19 (13.2%)	22 (16.8%)	0.40
LPV/r	17 (11.8%)	18 (13.7%)	0.63
CPE score	7.0 (6.0–7.2)	7.0 (6.0–8.0)	0.96
CSF CXCL10, pg/mL	1568.8 (947.5–3508.8)	1897.2 (1141.0–3465.9)	0.69
CSF TNFα, pg/mL	0.36 (0.29–0.49)	0.44 (0.33–0.59)	0.18
CSF IL-6, pg/mL	3.21 (2.43–4.29)	3.34 (2.20–4.65)	0.99

Legend: HCV = hepatitis C virus; ART = combination antiretroviral therapy; FTC, emtricitabine; TDF, tenofovir disoproxil; 3TC, lamivudine; ZDV, zidovudine; ABV, abacavir; DDI, didanosine; D4T, stavudine; ARV, antiretroviral; EFV, efavirenza; ATV/r, atazanavir/ritonavir; NVP, nevirapine; LPV/r, lopinavir/ritonavir; CPE = central nervous system penetration efficacy; CSF = cerebrospinal fluid; CXCL10 = C-X-C Motif Chemokine Ligand 10; TNF-α = Tumor Necrosis Factor alpha; IL-6, Interleukin 6.

**Table 2 viruses-17-00749-t002:** Multivariable linear regression models for CSF CXCL10 and CSF TNF-α.

	CSF CXCL10 (n = 275)		CSF TNF-α (n = 275)	
	Model 1 (R^2^ = 0.16, *p* < 0.001) aβ (BCa 95%CI), *p*	Model 2 (R^2^ = 0.15, *p* < 0.001) aβ (95%CI), *p*	Model 1 (R^2^ = 0.07, *p* < 0.001) aβ (BCa 95%CI), *p*	Model 2 (R^2^ = 0.08, *p* = 0.004) aβ (95%CI), *p*
CPE value	−0.18 (−0.28; −0.072), *p* < 0.001	−0.20 (−0.29; −0.11) *p* < 0.001	−0.074 (−0.13; −0.015), *p* = 0.014	−0.075 (−0.14; −0.005), *p* = 0.036
Age, years	-	0.007 (−0.008; 0.022), *p* = 0.352	-	0.004 (−0.008; 0.015), *p* = 0.536
Male sex, ref. female	0.38 (0.080; 0.68) *p* = 0.015	0.41 (0.086; 0.72), *p* = 0.013	-	0.062 (−0.18; 0.30), *p* = 0.616
White race, ref. others	-	0.017 (−0.24; 0.27), *p* = 0.895	-	−0.078 (−0.27; 0.12), *p* = 0.433
CD4/CD8 ratio	−0.039 (−0.47; 0.48), *p* = 0.863	Excluded	-	-
Nadir CD4+ T cell count, cells/µL	−0.11 (−0.38; 0.15) *p* = 0.424	Excluded	-	-
AIDS episode, ref. none	0.16 (−0.14; 0.50), *p* = 0.304	0.26 (0.004; 0.52), *p* = 0.047	0.18 (0.013; 0.36), *p* = 0.045	0.19 (−0.007; 0.39), *p* = 0.059
PI use, ref. no use	0.24 (−0.31; 0.83), *p* = 0.360	Excluded	-	-
NNRTI use, ref. no use	0.12 (−0.39; 0.72), *p* = 0.636	Excluded	-	-
Duration of current ART regimen, months	−0.33 (−0.58; −0.086), *p* = 0.010	−0.35 (−0.60; −0.11), *p* = 0.005	−0.25 (−0.44; −0.075), *p* = 0.005	−0.27 (−0.45; −0.086), *p* = 0.004

Model 1 included all covariates with significant associations in univariable analysis (*p* < 0.05) and entry method; model 1 underwent bootstrapping using 1000 samples (the shown aβ, 95%CI, and *p* values are bias-corrected and accelerated). Model 2 included all covariates with significant associations in univariable analysis (*p* < 0.05), and age, sex, race, and duration of ART regardless of univariable association; then, backward selection was applied based on the Akaike information criterion. Variables labeled as “Excluded” were entered into the model but not retained after backward selection. As such, beta coefficients and *p*-values are not reported. Legend: aβ = adjusted beta coefficient; BCa, bias-corrected and accelerated; 95%CI, 95% confidence interval; CPE = central nervous system penetration efficacy; CXCL10 = C-X-C Motif Chemokine Ligand 10; CSF = cerebrospinal fluid; TNF-α = Tumor Necrosis Factor alpha; PI = protease inhibitor; NNRTI = non-nucleoside reverse transcriptase inhibitor; ART = combination antiretroviral therapy.

## Data Availability

The data that support the findings of this study are available from the corresponding author, S.L.L., upon reasonable request.

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
