# Peer review of "Relationship Between Estimated Drug Distribution of Antiretroviral Therapy and Immune Proteins in Cerebrospinal Fluid During Chronic HIV Suppression"

_viruses, 2025, doi:10.3390/v17060749_

Round 1
Reviewer 1 Report
Comments and Suggestions for Authors
We know that higher CPE is associated with more effective viral suppression in the CNS. It is not known whether higher CPE scores are associated with neuroinflammation. The authors sought to fill this gap by evaluating the association of CPE score and three known CSF biomarkers of inflammation. Thus, the goal is to get the best estimate of the association while accounting for potential confounders.
The approach used is overly reliant on p values and does not use appropriate epidemiological methods. Reasons to include a variable are to control for confounding, to decrease extraneous error in the exposure, or to decrease extraneous error in the outcome. Does adding/removing a variable change the value of beta or change the standard error? P-values are not the primary concern in determining whether to include a variable. Including unnecessary variables can reduce precision of the estimate.
What other things are associated with neuroinflammation? Perhaps smoking?
Functional form of continuous variables. It appears that all variables were treated as linear predictors. We can see from the plots in Figure 1 that the association of CPE with biomarkers is approximately linear. What was done to assure that the association of other covariates (e.g. age, BMI, CD4) is linear? The association might be linear, but it could also be S-shaped, U-shaped, or J-shaped. This does not need to be shown in the paper, but it is necessary to do this and report findings.
Table 1 should include betas and confidence intervals for univariate and multivariate. IL-6 should also be included. Summary statistics for all variables could be included in this table as well. For example, we are told that the mean age is 43.8 and that the median CD4 was >500 but we are not told the distribution. If there is little to no variation in a covariate there is no point in including in a multivariable model.
Please also report when lumbar puncture samples were obtained.
Author Response
Reviewer#1:
We know that higher CPE is associated with more effective viral suppression in the CNS. It is not known whether higher CPE scores are associated with neuroinflammation. The authors sought to fill this gap by evaluating the association of CPE score and three known CSF biomarkers of inflammation. Thus, the goal is to get the best estimate of the association while accounting for potential confounders.
The approach used is overly reliant on p values and does not use appropriate epidemiological methods. Reasons to include a variable are to control for confounding, to decrease extraneous error in the exposure, or to decrease extraneous error in the outcome. Does adding/removing a variable change the value of beta or change the standard error? P-values are not the primary concern in determining whether to include a variable. Including unnecessary variables can reduce precision of the estimate.
What other things are associated with neuroinflammation? Perhaps smoking?
AR: We thank the reviewer for this thoughtful comment, which provided a valuable opportunity to clarify our methodological rationale. We agree that variable selection in regression models should be primarily driven by epidemiological reasoning and biological plausibility, rather than solely by statistical significance; and these principles informed our approach throughout.
The selection of covariates was informed by both epidemiological/biological relevance and data-driven considerations. Specifically, age, sex, race and duration of ART were retained in all models regardless of their statistical significance at univariate analysis because of their established relevance as potential confounders in the context of inflammation. Other variables were evaluated for inclusion through univariate based on previously reported biological associations with CSF inflammation in people with HIV, including nadir and current CD4+ counts, duration of HIV infection, prior AIDS-defining events, and thereafter based on p values. This p value filter was needed given the constraints posed by our sample size (n=275), which required balancing comprehensiveness with model stability and avoiding overfitting and collinearity (e.g., CD4 nadir and AIDS episode). Furthermore, to refine our estimates while minimizing risk of overfitting and multicollinearity, we used and showed two complementary modeling strategies: Model 1) Included all covariates with p<0.05 in univariate analysis (more p-value oriented). Model 2) Added biologically relevant covariates (age, sex, ART duration regardless of p values) and employed backward selection using the Akaike Information Criterion for optimal model fit (more biology driven).
Additionally, all final models were bootstrapped over 1,000 samples to improve internal validity and assess robustness to sampling variability: the CPE–biomarker associations remain consistent across bootstrap samples, even if the covariate structure slightly changes, which is also a way to address the question "Does adding/removing a variable change the value of beta or its precision?".
Regarding the reviewer’s suggestion to assess the impact of covariate inclusion on effect estimates, we observed mild changes in the beta coefficients for the association between CPE and CSF biomarkers across models. The betas and 95%CI for univariate and multivariate models are now presented in the new Table 2 and in the new supplementary Table 1. The relatively modest change in the primary association between CPE and inflammation markers between models’ approaches and after adjustment suggests that the selected covariates helped refine the estimates (as reflected also by slight improvements in R² values). This consistency supports the robustness of our findings and was part of our rationale for presenting two modeling strategies.
In line with the reviewer’s concern about overreliance on statistical significance, we have focused on effect sizes and consistency across training and validation groups, rather than solely p values, and we have acknowledged that the effect estimates of CPE on the biomarkers, despite significant, were small (lines 260-264 “While the effect size of such associations was small, the values were strikingly similar in the TG and VG, and confirmed in adjusted models computed in 1,000 random sub-samples; moreover, while covariates other than CPE were independently associated with CSF CXCL10 and TNF-α, including sex and AIDS diagnosis, their effect size was similar to that of CPE. These findings suggest that inflammation in CSF in PWH is at least partially dependent on CSF ART drug concentrations”).
Lastly, we appreciate the suggestion to consider additional potential confounders such as smoking, but smoking status is not available in our dataset, as well as other factors that may be associated with CSF inflammation, and we acknowledged several of these in the discussion (lines 322-327 “Even though we adjusted for the duration of ART, sex, and age, we likely did not fully account for all confounding factors that might influence neuroinflammation; among these, co-infections other than HCV [66,67], genetic predisposition [68,69], comorbidities and co-medications (e.g., antidepressants, statins) [57,70,71], and interindividual differences in ART metabolism and HIV reservoirs [72,73]” and following). In this regard, and in line also with a comment of Reviewer 2, we have better specified that exclusion criteria for neurological disorders included CNS cancers, head trauma, meningitis, autoimmune diseases, epilepsy, and psychiatric conditions other than anxiety and depressive disorders (e.g., schizophrenia). These exclusions were intended to minimize the influence of unrelated neurological comorbidities and reduce residual confounding in our analyses (lines 95-100 “Participants with untreated systemic or CNS infections, or other potentially confounding neurological conditions (e.g., stroke, head trauma, CNS cancers, CNS autoimmune diseases, epilepsy, and psychiatric disorders other than anxiety and depression, such as schizophrenia) were excluded to reduce the impact of unrelated neurological comorbidities. These exclusions limit generalizability but strengthen the internal validity of our findings by allowing a more specific focus on neuroinflammation in the context of HIV and ART”).
Functional form of continuous variables. It appears that all variables were treated as linear predictors. We can see from the plots in Figure 1 that the association of CPE with biomarkers is approximately linear. What was done to assure that the association of other covariates (e.g. age, BMI, CD4) is linear? The association might be linear, but it could also be S-shaped, U-shaped, or J-shaped. This does not need to be shown in the paper, but it is necessary to do this and report findings.
AR: Thanks for this methodological comment. We routinely assess the variables included in a study by visually inspecting scatterplots and LOESS curves for each variable of interest (in this case, CD4 counts, BMI etc) against the outcome variable (CPE in this case). For deeper investigations, we fit fractional polynomial regression or cubic splines. Age, CD4 counts, durations of ART regimens (current and total), and BMI were all adequately modeled as linear predictors based on this assessment. We have now reported in the methods, lines 128-130 “To assess whether continuous covariates were appropriately modeled as linear predictors, we applied multivariable fractional polynomial regression and assess LOESS curves.” We are thankful to the reviewer for asking about this and providing the opportunity to review the data as we noticed that some continuous covariates (e.g., CD4 counts) were not log-transformed, as for other continuous variables. We have now corrected this, despite no relevant changes in the results/findings of the linear regressions.
Table 1 should include betas and confidence intervals for univariate and multivariate. IL-6 should also be included. Summary statistics for all variables could be included in this table as well. For example, we are told that the mean age is 43.8 and that the median CD4 was >500 but we are not told the distribution. If there is little to no variation in a covariate there is no point in including in a multivariable model.
AR: We thank the reviewer for these suggestions. In response, we have updated the original Table 1 now Table 2 to include beta coefficients along with their 95% confidence intervals, as requested, and created Supplementary Table 1 for univariate analysis with betas and 95%CI. Regarding IL-6, we opted not to include it in this table to preserve readability and formatting. Presenting results for all three biomarkers would require a multipage layout, reducing clarity, and as uni- and multivariable analyses for IL-6 yielded no meaningful associations, we reported the negative findings in the text only and, for completeness, we presented the univariate analysis for IL-6 in supplementary Table 1 and 2 (this last presents additional covariates explored in response to Reviewer#2’s comments on single antiretroviral drugs).
To address the comment on descriptive statistics, we have added a new Table 1 presenting comprehensive summary statistics for all covariates. This provides readers with a detailed view of variable distributions and supports the assessment of covariate variability.
Please also report when lumbar puncture samples were obtained
AR: We have now specified in the methods “CSF samples were collected between 2003 and 2008 via morning spinal taps to minimize diurnal variability in CSF concentrations”.
We appreciate the reviewer’s careful reading and constructive input.

Reviewer 2 Report
Comments and Suggestions for Authors
Summary:
In the presented article titled “Relationships Between Estimated Drug Distribution of Antiretroviral Therapy and Immune Proteins in Cerebrospinal Fluid during Chronic HIV Suppression”, Trunfio et al. carry out a comparative analysis of central nervous system (CNS) penetration effectiveness of various HIV antiretroviral therapy (ART) regimens in relation to inflammation biomarkers (CXCL10, TNF-alpha, and IL-6) in cerebrospinal fluid. The authors investigated a total of n=275 patient samples from the CNS HIV Antiretroviral Therapy Effects Research (CHARTER) study, divided into a training and validation cohort. They found that treatment regimens with a higher CNS penetration effectiveness (CPE) score were associated with lower inflammatory markers (CXCL10 and TNF-alpha). They conclude that distinct ART regimens may produce different neuroimmune/inflammatory signatures and may potentially be linked to neurocognitive and psychiatric outcomes in people living with HIV.
General concept comments:
The authors emphasize that CPE is a useful correlate for CNS control for HIV, but stress that its association with neuroinflammation has not been addressed. They further highlight conflicting findings in clinical trials regarding CPE and neurocognitive outcomes. Given the pleiotropic effects of many ART drugs on neuronal function, the study addresses and important aspect in their use, namely if the CNS penetration of ART regimens is associated with distinct neuroinflammatory profiles. This topic is of growing significance, given the increasing age and longevity of many people living with HIV, and their long-term CNS exposure to these drugs. The effects of ART regimens beyond viral control are thus of increasing importance.
The article references numerous (clinical) studies from recent years (2020-2025) highlighting the status of the field and gaps in knowledge, as well as conflicting findings from previous clinical studies. The methodology is laid out mostly clearly, and the choice of biomarker readouts (CXCL10, TNF-alpha, and IL-6) is well justified and described based on their elevated levels under ART and previously reported association with cognitive endpoints in HIV patients. Statistical analysis is described and summarized in two tables and one figure. Multivariate analysis is performed in the combined sample set and confirmed the observed associations.
The discussion is detailed, insightful and cross-references multiple related literature findings. The authors acknowledge shortcomings of their study, especially when it comes to establishing causality in the observed metrics and with regard to longitudinal data readouts.
While overall clear, the methodology is at times not described in enough detail to fully recapitulate data analysis and should be described in more detail as outlined below. Some of the distinct antiviral drug regimens should be addressed individually, especially given their different modes of action and possible impact on inflammatory patterns.
Another important aspect in the presented research is confounding by CNS co-medications/co-morbidities. While some co-morbidities were excluded, this approach cannot fully mitigate the possible impact of common co-morbidities like depression or psychotic disorders. Ideally, the authors may clarify co-medications/morbidities in more detail if data are available.
Specific comments
- Line 95/96: The authors state that certain types of CNS disorders (CNS infections, head trauma) were excluded. However, other potential co-morbidities and co-medications may confound the findings. If available, the authors should comment on or summarize co-medications with CNS-active drugs (e.g. antidepressants, antipsychotics, anticonvulsants), and related co-morbidities. Such co-morbidities should be assessed as a confounding factor as well.
- The authors comment on (line 149-152) and discuss (line 286-305) different ART drug regimens, but do not perform an analysis to assess if there are significant differences between common ART regimens in their study cohort. It would help put the presented work into context if the authors listed CXCL10 and TNF-alpha levels with the most common ART combinations (e.g. emtricitabine+tenofovir, lamivudine+zidovudine). While CPE scores are obviously key differences between these regimens, it stands to reason that there could be differences among drugs/drug classes that are not associated with CPE score.
- Line 126/127: The authors state that they accounted for potential confounding factors, but do not explicitly specify which factors they corrected for. This information should be added to the methods section.
- Line 235: ”IL-6 may be a better marker of legacy effects […]”. The wording here is not fully clear. The authors should describe briefly which effects they are referring to.
- Table 1: It is not clear under which circumstances Model 2 is excluded, and what the respective p-values and beta coefficients would have been. This needs to be clarified.
- Table 1: for transparency, the authors should state effect sizes in pg/ml or log(pg/ml) and also specify in more detail how the standardized adjusted beta coefficient is defined (i.e. describe in the methods section).
Author Response
Reviewer#2
In the presented article titled “Relationships Between Estimated Drug Distribution of Antiretroviral Therapy and Immune Proteins in Cerebrospinal Fluid during Chronic HIV Suppression”, Trunfio et al. carry out a comparative analysis of central nervous system (CNS) penetration effectiveness of various HIV antiretroviral therapy (ART) regimens in relation to inflammation biomarkers (CXCL10, TNF-alpha, and IL-6) in cerebrospinal fluid. The authors investigated a total of n=275 patient samples from the CNS HIV Antiretroviral Therapy Effects Research (CHARTER) study, divided into a training and validation cohort. They found that treatment regimens with a higher CNS penetration effectiveness (CPE) score were associated with lower inflammatory markers (CXCL10 and TNF-alpha). They conclude that distinct ART regimens may produce different neuroimmune/inflammatory signatures and may potentially be linked to neurocognitive and psychiatric outcomes in people living with HIV.
General concept comments:
The authors emphasize that CPE is a useful correlate for CNS control for HIV, but stress that its association with neuroinflammation has not been addressed. They further highlight conflicting findings in clinical trials regarding CPE and neurocognitive outcomes. Given the pleiotropic effects of many ART drugs on neuronal function, the study addresses and important aspect in their use, namely if the CNS penetration of ART regimens is associated with distinct neuroinflammatory profiles. This topic is of growing significance, given the increasing age and longevity of many people living with HIV, and their long-term CNS exposure to these drugs. The effects of ART regimens beyond viral control are thus of increasing importance.
The article references numerous (clinical) studies from recent years (2020-2025) highlighting the status of the field and gaps in knowledge, as well as conflicting findings from previous clinical studies. The methodology is laid out mostly clearly, and the choice of biomarker readouts (CXCL10, TNF-alpha, and IL-6) is well justified and described based on their elevated levels under ART and previously reported association with cognitive endpoints in HIV patients. Statistical analysis is described and summarized in two tables and one figure. Multivariate analysis is performed in the combined sample set and confirmed the observed associations.
The discussion is detailed, insightful and cross-references multiple related literature findings. The authors acknowledge shortcomings of their study, especially when it comes to establishing causality in the observed metrics and with regard to longitudinal data readouts.
While overall clear, the methodology is at times not described in enough detail to fully recapitulate data analysis and should be described in more detail as outlined below. Some of the distinct antiviral drug regimens should be addressed individually, especially given their different modes of action and possible impact on inflammatory patterns. Another important aspect in the presented research is confounding by CNS co-medications/co-morbidities. While some co-morbidities were excluded, this approach cannot fully mitigate the possible impact of common co-morbidities like depression or psychotic disorders. Ideally, the authors may clarify co-medications/morbidities in more detail if data are available.
Specific comments
- Line 95/96: The authors state that certain types of CNS disorders (CNS infections, head trauma) were excluded. However, other potential co-morbidities and co-medications may confound the findings. If available, the authors should comment on or summarize co-medications with CNS-active drugs (e.g. antidepressants, antipsychotics, anticonvulsants), and related co-morbidities. Such co-morbidities should be assessed as a confounding factor as well.
AR: We are thankful for this comment, as a research group we are very interested in this aspect. Unfortunately, we do not have enough data on co-medications (limited range of medications assessed with missing data for more than half the study population). The study population is a subsample within the first old cohort of participants enrolled in CHARTER and several variables were not recorded during the initial years of enrollment (2003/2008). We have acknowledged this limitation (lines 322-326 “Even though we adjusted for the duration of ART, sex, race, and age, we likely did not fully account for all confounding factors that might influence neuroinflammation; among these, co-infections other than HCV [66,67], genetic predisposition [68,69], comorbidities and co-medications (e.g., antidepressants, statins) [57,70,71], and interindividual differences in ART metabolism and HIV reservoirs [72,73]”).
Furthermore, we have clarified that in selecting participants to focus specifically on HIV suppression/CPE-related effects on CSF inflammation, we excluded individuals with neurological conditions and psychiatric disorders other than anxiety and depression (e.g., schizophrenia). As a result, none of the participants were taking neuroleptics, antiepileptics, or similar medications (lines 95-100 “Participants with untreated systemic or CNS infections, or other potentially confounding neurological conditions (e.g., stroke, head trauma, CNS cancers, CNS autoimmune diseases, epilepsy, and psychiatric disorders other than anxiety and depression, such as schizophrenia) were excluded to reduce the impact of unrelated neurological comorbidities. These exclusions limit generalizability but strengthen the internal validity of our findings by allowing a more specific focus on neuroinflammation in the context of HIV and ART”).
- The authors comment on (line 149-152) and discuss (line 286-305) different ART drug regimens, but do not perform an analysis to assess if there are significant differences between common ART regimens in their study cohort. It would help put the presented work into context if the authors listed CXCL10 and TNF-alpha levels with the most common ART combinations (e.g. emtricitabine+tenofovir, lamivudine+zidovudine). While CPE scores are obviously key differences between these regimens, it stands to reason that there could be differences among drugs/drug classes that are not associated with CPE score.
AR: We thank the reviewer for raising this important point. In response, we have explored the most common backbones, and third agents used in our cohort, shown the comparisons between TG and VG in new Table 1 (no differences), and provided supplementary models (univariable and multivariable linear regressions in supplementary Table 1, 2, and 3) that included single antiretroviral drugs. Less frequently used antiretrovirals (fosamprenavir, indinavir, nelfinavir, saquinavir) were not assessed due to sample sizes <10, which would result in highly unstable estimates.
Before commenting on these additional findings, we want to acknowledge the following: our study was not designed or powered to investigate the independent effects of specific antiretroviral drugs. The number of participants per drug is insufficient to allow for robust comparisons, especially when considering adjustment for confounders. Furthermore, many antiretrovirals are highly collinear with the CPE score (as CPE is a composite score derived from ARV composition), making it statistically problematic to include both in the same model. The risk of overadjustment and collinearity reduces interpretability and undermines the validity of any drug-specific inference, as well as of CPE (variance inflation factor between CPE and specific ARVs was up to 4 in some models, and up to 8 in models that included antiretroviral classes -PIs, NNRTIs- as covariates). Additionally, investigating multiple ARVs inflates the risk of type I errors, and lastly, from a biological and methodological perspective, we did compare a single antiretroviral -e.g. ATV - to a reference group represented by all participants not on that ARV. If the research question is to investigate immune-modulating/inflammatory activity of ATV (or of any other ARV), this reference group is not appropriate, as it is a pooled group mixing many ARVs with potentially dissimilar activity (some could be similar to ATV, some could have no effects, and some could have opposite effects to ATV), thereby diluting interpretability; eventually, in case of a significant association, it is hard to understand what biological phenomenon is detected and in case of a negative finding, we are not confident enough (due to all the above) in suggesting that the drug has no impact on CSF inflammation. The effects of the third ARV may also depend on the accompanying NRTI backbone, which further complicates interpretation, as there were 8 more combinations and 14 participants in 3-classes regimens, other than those on the 7 most common backbones shown in Table 1. Attempts to include the most common backbones in the models do not fully resolve this issue, given the multiplicity of combinations, relatively small numbers in each category, and further collinearity not only with CPE but with the single antiretrovirals (as specific backbones are more frequently combined with specific third drugs).
For all these reasons, we explored the associations between the most common antiretrovirals and CSF biomarkers
-through bivariate analysis in Supplementary Table 2: what is interesting here, other than single associations, is that 2 out of the 3 ARVs with the lowest CPE (1-2) were significantly associated with higher CXCL10 and TNFa, while 2 out of the 2 ARVs with the highest CPE (4) were significantly associated with lower biomarkers. In the group of ARVs with CPE=3, only LPV was significantly associated with lower TNFa. This may suggest a pattern of effect again dependent on the penetration of the drugs into the CNS rather than antiretroviral-specific effect.
-through multivariable models in Supplementary Table 3 (for CXCL10) and 4 (for TNFa): we explored the effects of including univariate significant ARVs (from the previous analysis) in the main models 1 and 2 of Table 2. For these, we also assessed the effect of removing CPE score from the models, leaving only ARVs. Therefore, supplementary Table 3 and 4 show 2 Models 1 (with and without CPE), and 2 Models 2 (with and without CPE). For Suppl.Table 3, CPE confirmed its association with CSF CXCL10 independently from single ARVs, and ATV and TDF use showed an independent association with higher CXCL10 that was statistically significant only when CPE was not included. Collinearity (VIFs) in the final models was tolerable (<4), but models not including CPE had lower R2. For Suppl.Table 4, CPE and all 5 included ARVs were not significantly associated with TNFa (VIFs up to 4,5), while removing CPE led to significant associations between TNFa and either ATV or both ATV and LPV depending on the models.
For all the reasons above (limited power, multicollinearity, risk of overadjustment, biological ambiguity, and assessment of changes across models with or without CPE and ARVs), we believe that drawing strong conclusions on individual drug-level analyses, beyond descriptive exploration, are not methodologically appropriate in this study and dataset (a meaningful investigation of this question would require a study explicitly designed for that purpose, with a larger, more homogeneous sample and randomized or matched assignment of regimens). We do agree with the reviewer that differences in ARV pharmacodynamics and off-target effects are biologically plausible, so we have introduced these paragraphs in the manuscript:
-Results section: “To further investigate the relationship between individual ARVs and CSF inflammation, we conducted exploratory analyses based on the most commonly used ARVs, reported in Table 1. Univariable associations between single ARVs and CSF biomarkers (Supplementary Table 2) revealed that drugs with lower CPE ranking (tenofovir disoproxil and atazanavir, both CPE ≤2) were associated with higher levels of CSF CXCL10 and TNF-α. Conversely, ARVs with the highest CPE (zidovudine and nevirapine, both CPE = 4) were associated with lower levels of these biomarkers. Among drugs with intermediate CPE (CPE = 3), only lopinavir showed a significant association with lower TNF-α levels (Supplementary Table 2).
In multivariable models for CSF CXCL10 (Supplementary Table 3), CPE remained independently associated with lower biomarker levels, even after inclusion of individual ARVs. Notably, atazanavir and tenofovir disoproxil were associated with higher CXCL10 only in models excluding CPE. For CSF TNF-α (Supplementary Table 4), CPE and all five univariable-selected ARVs were not significantly associated in full models. However, when CPE was excluded, atazanavir showed significant associations with higher TNF-α levels and lopinavir with lower biomarker levels, while backward selection retained only atazanavir as associated with higher TNF-α levels (Supplementary Table 4). Across all models, collinearity was detected between CPE and ARVs (VIFs ranging from 1.2 to 4.5), and models excluding CPE explained mildly less variance (lower R²).”
-Discussion: “In line with this, we conducted exploratory analyses of the single ARVs most commonly used in our cohort to assess whether specific agents might be independently associated with CSF inflammation. In univariable models, drugs with lower CPE were associated with higher levels of CXCL10 and TNF-α, while those with the highest CPE score were associated with lower biomarker levels. This pattern suggests a potential gradient of effect, indicating that CNS inflammation may be more closely tied to drug penetration than to molecule-specific effects. In multivariable models, CPE remained significantly associated with CXCL10 even after adjusting for individual ARVs. The inclusion or exclusion of CPE influenced the associations of atazanavir and tenofovir with CXCL10, with both ARVs associated with higher CSF levels when CPE was excluded. For TNF-α, the significance of CPE, atazanavir, and lopinavir varied depending on whether they were included together in the models. These findings are exploratory and should be interpreted cautiously as limited by sample size, model complexity, collinearity issues, heterogeneous comparators, and shared explanatory variance.”
- Line 126/127: The authors state that they accounted for potential confounding factors, but do not explicitly specify which factors they corrected for. This information should be added to the methods section.
AR: Apologies, we have now better described the approach and methods for multivariable modeling, as follow (lines 134-146) “Multivariable linear regression models were used to assess the independent associations between CPE scores and CSF biomarker levels. Two model-building strategies were employed, and were run in the whole study population to increase the power and allow multiple adjustments. Models 1 included all covariates with significant univariable associations (p<0.05) and retained them without further selection; to ensure robustness and maintain the internal validation approach, these models were also bootstrapped using 1,000 random samples. Models 2 included the same univariable-significant covariates along with the predefined confounders (age, sex, race, and ART duration), and then applied backward selection based on the Akaike Information Criterion (AIC). Due to the nature of backward selection, bootstrapping was not applied to Models 2. Biomarker values were log₁₀-transformed prior to modeling, and regression coefficients are reported on the log₁₀-transformed scale (i.e., log₁₀ pg/mL). Exploratory analyses on the associations between single ARVs and CSF biomarkers levels were also performed with the same modeling approach”.
- Line 235: ”IL-6 may be a better marker of legacy effects […]”. The wording here is not fully clear. The authors should describe briefly which effects they are referring to.
AR: We appreciate the opportunity to clarify this sentence. By "legacy effects," we refer to persistent immunological alterations or neurological injury resulting from earlier phases of uncontrolled viremia, delayed ART initiation, or prior neuroinflammatory insults that are not reversed by current ART regimens and viral suppression. Based on prior literature (persistently altered IL-6 levels despite viral suppression after years, compared to people without HIV), and our finding, IL-6 may reflect these longer-lasting, cumulative and residual effects rather than ongoing viral activity. We now modified the sentence as (lines 295-301): “Overall, IL-6 may be a better marker of legacy effects (meant as, residual immunological alterations and CNS tissue injury established before ART initiation and that persist despite long-term viral suppression[54]). As such, IL-6 may reflect cumulative inflammatory burden rather than being a sensitive indicator of current or residual viral activity, explaining our negative findings, as well as previous observations [51–53]"”
- Table 1: It is not clear under which circumstances Model 2 is excluded, and what the respective p-values and beta coefficients would have been. This needs to be clarified.
AR: We thank the reviewer for this important observation. In our analysis, Model 2 was computed including all variables included in Models 1; however, some covariates were not retained (=”Excluded”) in the final Model 2 due to the backward selection process based on the Akaike Information Criterion. So Model 1 was run with entry method: all the variables are retained in the final model. Models2 used backward selection: only some variables are retained in the final model, based on AIC which leads to the exclusion of certain covariates during the model-fitting process. For those covariates, beta coefficients and confidence intervals cannot be reported because they do not belong to the final selected model (they would be those from the last iteration that included that variable before dropping it off). Reporting estimates for these variables would reflect intermediate, non-final models and would not correspond to the optimized model results. We have clarified this point in the Tables legend and in the Methods section.
Legend of Tables “Models 1 were performed including all covariates with significant associations at univariable analysis (p<0.05) and entry method; models 1 underwent bootstrapping in 1,000 samples (the shown aβ, 95%CI, and p values are bias corrected and accelerated). Models 2 were performed including all covariates with significant associations at univariable analysis (p<0.05) and age, sex, race, and duration of ART regardless of univariable association; then backward selection was applied based on the Akaike information criterion. Variables labeled as "Excluded" were entered into the model but not retained after backward selection. As such, beta coefficients and p-values are not reported”
Methods section, lines 134-146: “Multivariable linear regression models were used to assess the independent associations between CPE scores and CSF biomarker levels. Two model-building strategies were employed, and were run in the whole study population to increase the power and allow multiple adjustments. Models 1 included all covariates with significant univariable associations (p<0.05) and retained them without further selection; to ensure robustness and maintain the internal validation approach, these models were also bootstrapped using 1,000 random samples. Models 2 included the same univariable-significant covariates along with the predefined confounders (age, sex, race, and ART duration), and then applied backward selection based on the Akaike Information Criterion (AIC). Due to the nature of backward selection, bootstrapping was not applied to Models 2. Biomarker values were log₁₀-transformed prior to modeling, and regression coefficients are reported on the log₁₀-transformed scale (i.e., log₁₀ pg/mL). Exploratory analyses on the associations between single ARVs and CSF biomarkers levels were also performed with the same modeling approach”
- Table 1: for transparency, the authors should state effect sizes in pg/ml or log(pg/ml) and also specify in more detail how the standardized adjusted beta coefficient is defined (i.e. describe in the methods section)
AR: We thank the reviewer for their time spent improving our manuscript. In response, we have revised the tables to report beta coefficients and 95% confidence intervals for both univariable (supplementary table 1 and 2) and multivariable models (new table 2, and supplementary tables 3-4). These estimates reflect the log10-transformed pg/mL values of the biomarkers, which align with how the outcome variables were modeled and improves interpretability; and we have now specified this in the methods (as reported in the previous answers). We used only unstandardized beta coefficients to avoid confusion between standardization from bootstrapped and not bootstrapped results.

Round 2
Reviewer 1 Report
Comments and Suggestions for Authors
I commend you on a comprehensive revision.
Reviewer 2 Report
Comments and Suggestions for Authors
In the presented revised article titled “Relationships Between Estimated Drug Distribution of Antiretroviral Therapy and Immune Proteins in Cerebrospinal Fluid during Chronic HIV Suppression”, Trunfio et al. carry out a comparative analysis of central nervous system (CNS) penetration effectiveness of various HIV antiretroviral therapy (ART) regimens in relation to inflammation biomarkers (CXCL10, TNF-alpha, and IL-6) in cerebrospinal fluid. The authors investigated a total of n=275 patient samples from the CNS HIV Antiretroviral Therapy Effects Research (CHARTER) study, divided into a training and validation cohort. They found that treatment regimens with a higher CNS penetration effectiveness (CPE) score were associated with lower inflammatory markers (CXCL10 and TNF-alpha). They conclude that distinct ART regimens may produce different neuroimmune/inflammatory signatures and may potentially be linked to neurocognitive and psychiatric outcomes in people living with HIV.
The authors have addressed criticisms from reviewers in detail in a point-by-point rebuttal and also provided additional written sections in methods, results and discussion, as well as a new data table (table 1). Four new supplementary tables further strengthen the detail of the presented analysis. Overall, they highlight additional insights into individual ART regimens in their study population and co-medications/co-morbidities. The authors transparently highlight limitations of the study and collected data. They discuss them in additional depth.
Methods:
The methodology is laid out more clearly now, with reviewer criticisms taken into consideration. The methods section was expanded to provide more clarity, especially regarding the two-model approach that the authors present in table 2. The description of the two model-building strategies (model1/2) is now much more detailed. Beta coefficients and 95% confidence intervals for both univariable and multivariable models are now reported throughout the manuscript and supplementary material, thereby strengthening data transparency. Additional explanation of study exclusion criteria is given in the methods section. This clarifies the exclusion of patients with CNS conditions (e.g., stroke, head trauma, CNS cancers, CNS autoimmune diseases, epilepsy, schizophrenia). In line with the authors’ comments, this approach improves the internal validity of the study and removes potential confounding variables.
Results:
In the revised version of the manuscript, statistical analysis is described more thoroughly and summarized in a new table 1. Data are now presented more clearly in supplementary tables. Supplementary Tables 3 and 4 address key criticisms regarding the role of individual antiretroviral drug regimens. An additional section in the discussion addresses the conclusions from this analysis further.
Discussion:
Overall, the discussion and conclusions in the revised manuscript are now more detailed and nuanced. With regard to the additional data analysis included, the authors again discuss the limitations of such analysis. In particular, they discuss limitations regarding analysis of individual antiretroviral drug regimens, something the study is not fully powered to address. In their rebuttal, the authors further clarify this aspect.
In summary, the criticisms raised by the reviewer have been addressed in full, reflected in additional data analysis and discussed accordingly.